# Substrate Impact on the Structure and Electrocatalyst Properties of Molybdenum Disulfide for HER from Water

**Arūnas Jagminas \*, Arnas Naujokaitis, Paulius Gaigalas, Simonas Ramanavičius** 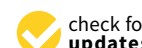**, Marija Kurtinaitienė and Romualdas Trusovas**

State Research Institute Center for Physical Sciences and Technology, Sauletekio ave. 3, LT-10257 Vilnius, Lithuania; arnas.naujokaitis@ftmc.lt (A.N.); paulius.gaigalas@ftmc.lt (P.G.); simonas.ramanavicius@ftmc.lt (S.R.); marija.kurtinaitiene@ftmc.lt (M.K.); romualdas.trusovas@ar.fi.lt (R.T.)

\* Correspondence: arunas.jagminas@ftmc.lt; Tel.:+370-5264-8891

**Abstract:** It is expected that utilization of molybdenum disulfide ($MoS_2$)-based nanostructured electrocatalysts might replace the Pt-group electrodes most effectively applied for hydrogen evolution reaction from water. Therefore, in the past two decades, various approaches have been reported for fabrication of nanostructured $MoS_2$-based catalysts, but their applications in practice are still missing due to lower activity and stability. We envisaged that the knowledge about the peculiarities of $MoS_2$ nanoplatelets attachment to various conductive substrates by hydrothermal processing could be helpful for fabrication of more active and stable working electrodes. Therefore, in this study, the hydrothermal syntheses at the Mo, Ti, Al, anodized Ti, and hydrothermally designed titanium suboxide substrates were performed; the electrodes obtained were characterized; and hydrogen evolution reaction (HER) activity was tested. In this way, $MoS_2$-based HER catalyst possessing a surprising stability and a low Tafel slope was designed via attachment of nanoplatelet-shaped $MoS_2$ species to the nanotube-shaped anatase-$TiO_2$ surface.

**Keywords:** electrocatalyst supports; hydrothermal synthesis; molybdenum disulfide; microstructure; hydrogen evolution

## 1. Introduction

Water splitting is the most promising way for hydrogen gas production attracting still growing attention in the past decade [1]. For this, besides bulk platinum and Pt alloys, as the most effective electrocatalysts for hydrogen evolution reaction (HER) from aqueous acidic solutions, application of various new nanostructured materials is currently of high interest [2–4]. Consequently, fabrication of new, cost-effective, and prospective electrocatalysts on the basis of nanostructured transition metal chalcogenides, molybdenum oxide [5], nitride [6], carbide [7], and phosphide [8] has been reported in the past decade. Noteworthy, if compared with platinum electrocatalysts, $MoS_2$-based ones were less effective and stable, but recently, they were classified as more extensively studied [9–11]. Gold [12], titanium [13], carbon paper [14], graphite [15], nickel foam [16], and graphene-based sheets [17,18] have been proposed as the supports for attachment of $MoS_2$-based nanostructures. However, the stability of reported nanostructured $MoS_2$ electrodes is not high enough to replace Pt group metals in the HER processes. Therefore, recent efforts have been focused on the developing more advanced $MoS_2$-based nanometer-scaled hybrid assembles using various approaches, including composition and morphology control [19], alloying [20], and support promotion [21]. A variety of conducting substrates such as reduced graphene oxide [22], carbon nanotubes [23], mesoporous graphene [24],

and carbon fibers [25] have been also reported seeking to enhance HER activity. For example, the design of film from metallic 1T-MoS$_2$ and semiconducting 2H-MoS$_2$ phases rendered the catalyst possessing a high HER activity [19,26] and an initial overvoltage as small as 40 mV [27]. Meanwhile, various synthesis approaches have been reported in the past years, including laser ablation [28], thermal decomposition [29], gas-phase reaction [30], magnetron sputtering [31], hydrothermal [24, 32–34], and sonochemical [35]. However, the design of stably working MoS$_2$-based electrocatalysts for hydrogen evolution at the conductive substrates is still challenging. Therefore, the creation of 2D-shaped MoS$_2$-based electrocatalyst well attached to conductive substrate is one of the fundamental problems that needs to be resolved.

This paper attempts to show the influence of how various substrates can be applied for the successive formation of nanoplatelet MoS$_2$ electrocatalysts by one-pot hydrothermal approach. To study the HER efficiency and stability, MoS$_2$ nanostructured films were designed onto the molybdenum, titanium, Si/SiO$_2$, and aluminum substrates. In addition, the influence of thin nanoporous anodic oxide films formed onto the titanium surface, underlying MoS$_2$, was investigated and discussed. In this way, the influence of the substrate nature on the HER activity stemmed from the quantity of exposed sulfur edges in the MoS$_2$-based film [12,36] was estimated.

## 2. Materials and Methods

### 2.1. Materials

For preparation of the synthesis solution, the ammonium heptamolybdate, (NH$_4$)$_6$Mo$_7$O$_{24}$ 4H$_2$O, was purchased from REACHEM (Petržalka, Slovakia), whereas thiourea and aniline were obtained from Sigma-Aldrich (St. Louis, Missouri, USA). All reagents were of analytical grade and used without further treatment. Deionized water purified in a Milli-Q system was used throughout. Mo (99.9 at%, 0.1 mm thick) and Ti (99.7 at%, 0.127 mm thick) foils, purchased from Sigma-Aldrich, were used to cut rectangular specimens with working surface of 1.0 cm$^2$ (7 × 7 mm$^2$) and tagged (1 × 30 mm$^2$). Al foil (99.99 at%, 0.13 mm thick) was purchased from Sigma-Aldrich. To clean sample surfaces, high purity acetone and ethanol were applied. For the preparation of anodizing solutions, chemically green H$_3$PO$_4$ and H$_2$SO$_4$ acids were purchased from Reachim (Moscow, Russia). For titanium suboxide, TiO$_x$, synthesis at the Ti substrate, selenious acid, H$_2$SeO$_3$, (the highest purity, purchased from Reachim) (Moscow, Russia) was applied.

### 2.2. Hydrothermal Synthesis of MoS$_2$ Species

For fabrication of flower-shaped MoS$_2$ species in the solution bulk and on various substrates, hydrothermal synthesis in the solution containing thiourea, (NH$_2$)$_2$CS, ammonium heptamolybdate and aniline, C$_6$H$_5$NH$_2$, was used. In these investigations, a Teflon-lined stainless-steel autoclave of 20 mL capacity was filled with the working solution to 60% and then sealed. The typical solution comprised 5.0 (NH$_4$)$_6$Mo$_7$O$_{24}$, 90 thiourea, and 25 mmol L$^{-1}$ aniline. The cleaned specimen was inserted vertically into the solution using a special holder made of Teflon. The synthesis reaction was carried out at 220 ± 2 °C for 5 to 15 h in a programmable muffle furnace Czermack (Badia Polesine, Italy) using 10 °C /min temperature ramp. As-formed products were cooled to room temperature, washed several times with distilled water, collected by centrifugation, and dried at 60 °C.

### 2.3. Ti Surface Pretreatment, Anodizing, and Oxidation

The surface of titanium samples was ultrasonically cleaned in acetone, ethanol, and water, 6 min in each, and air dried. Three kinds of Ti surface preparation were used in our subsequent study, namely, pure Ti, Ti covered by nanotube-shaped anatase TiO$_2$ layer via anodizing and calcination, and Ti covered with nanostructured titanium suboxide by hydrothermal treatment under conditions evaluated by us earlier [37]. Ti surface anodizing was conducted in the thermostated Teflon cell containing 2.0 mol L$^{-1}$ H$_3$PO$_4$ and 0.2 mol L$^{-1}$ NH$_4$F at 17 ± 0.3 °C and 20 V dc for 1 h. In this setup,

two platinum plates were used as cathodes. After anodizing, the specimens were thoroughly rinsed with water, air-dried, and calcined in air at 450 °C for 2 h using 10 °C min$^{-1}$ temperature ramp.

For Ti surface coating with the nanostructured titanium suboxide (TiO$_x$) film, the specimens were hydrothermally treated in 0.2 mol L$^{-1}$ H$_2$SeO$_3$ at 180 °C for 15 h, rinsed, and annealed in deoxygenated atmosphere by heating together with Cu foil in a sealed glass tube at 300 °C for 10 h, as reported earlier [37].

*2.4. Characterization Tests*

The morphology and elemental composition of the products obtained was investigated using a scanning electron microscope (FEI Helios Nanolab 650, Eindhoven, Niderland) and Cross Beam Workstation Auriga (Eindhoven, Nederland) equipped with a field emission gun and EDX spectrometer. A FEI TECNAI F20 (Eindhoven, Nederland) electron transmission microscope equipped with a field emission electron gun was also applied. Accelerating voltage was 200 kV. X-ray powder and glancing angle diffraction experiments were performed on a D8 diffractometer (Bruker AXS, Germany), equipped with a Göbel mirror as a primary beam monochromator for CuK$_\alpha$ radiation.

Raman spectra were recorded using inVia (Renishaw) spectrometer equipped with thermoelectrically cooled (−70 °C) CCD camera and microscope (New Mills, UK). The 532 nm beam of the solid-state laser was used as an excitation source. Raman scattering wavenumber axis was calibrated by the silicon peak at 520.7 nm. The 50× objective was used during the measurements of TiNT/MoS$_2$ and TiNT samples, while 20 × objective was employed for the analysis of MoS$_2$ in powder form. To avoid damage of the samples, the laser power was limited to 0.3 mW. Parameters of the bands were determined by fitting the experimental spectra with Gaussian–Lorentzian shape components using GRAMS/A1 (Thermo Scientific) software (version 8.0, Thermo Electron Corp.).

Electrochemical measurements were performed in a three-electrode glass cell equipped with the working, Ag/AgCl, KCl reference, and the glassy carbon electrode, as a counter. Prior to the measurements, Ar gas was bubbled through the electrolyte solution for at least 0.5 h to remove the dissolved air. The catalytic performance of prepared electrodes was studied in the solution containing 0.5 mol L$^{-1}$ H$_2$SO$_4$ and 0.25 mol L$^{-1}$ formic acid by the potential cycling within 0.05 to −0.35 V vs. reference hydrogen electrode RHE potentials range at the 10 mV s$^{-1}$ scan rate using a workstation ZENNIUM Zahner-Electrik Gmb & Co, KG, Kronach, Germany). Formic acid was added to hinder the destruction of glassy carbon counter electrode in our prolonged HER tests. In the stability tests, up to 3000 cycles were used. All potentials in the text refer to RHE.

## 3. Results

Structurally, the products, synthesized hydrothermally in the bulk solution containing ammonium heptamolybdate, thiourea, and aniline at 220 °C, are nanoplatelet-shaped flowers (Figure 1a). At the same time, densely packed nanoplatelet arrays usually covered various substrates inserted into the synthesis reactor (Figure 1b).

The thickness of individual nanoplatelet at the surface side varied between 3 and 7 nm (inset in Figure 1a) implying on the formation of 5–12 sandwiched S-Mo-S layers [19,38]. The thickness of films for the given solution was found to be dependent on the process duration varying from 700 to 800 nm and from 1.7 to 2.1 μm for 5 and 15 h, respectively. Note that the reaction times shorter than 3 h led to the formation of incompletely swelled coatings at the most tested substrates. When synthesis time lasted over 10 h, a substantial decrease in the working stability of such MoS$_2$-nanostructured electrocatalysts was observed under the intense HER. From the TEM observations, the high degree ordering in our MoS$_2$-based nanoplatelets is absent (Figure 2).

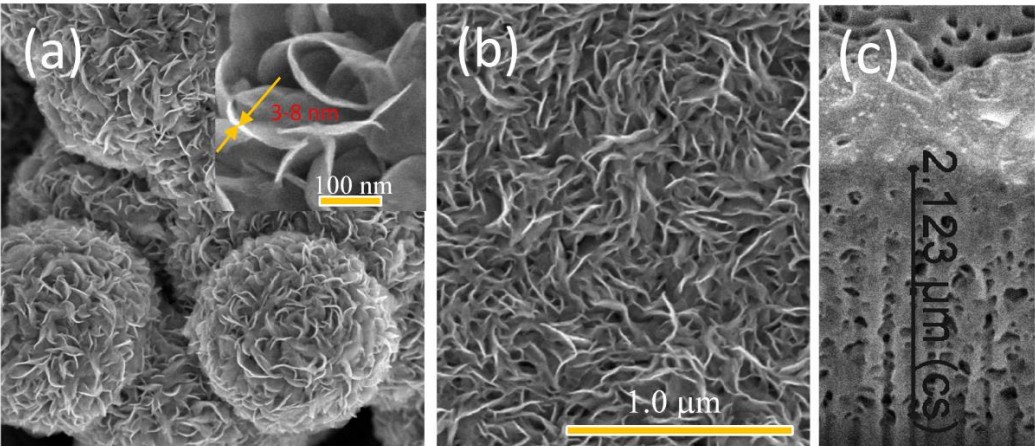

**Figure 1.** Typical top-side (**a**,**b**) and cross-sectional (**c**) SEM images of MoS$_2$ formed hydrothermally at 220 °C for 10 h in the solution containing (in mmol L$^{-1}$): 5 (NH$_4$)$_6$Mo$_7$O$_{24}$, 90 thiourea, and 25 aniline in bulk (**a**) and on the conducting substrate (**b**,**c**).

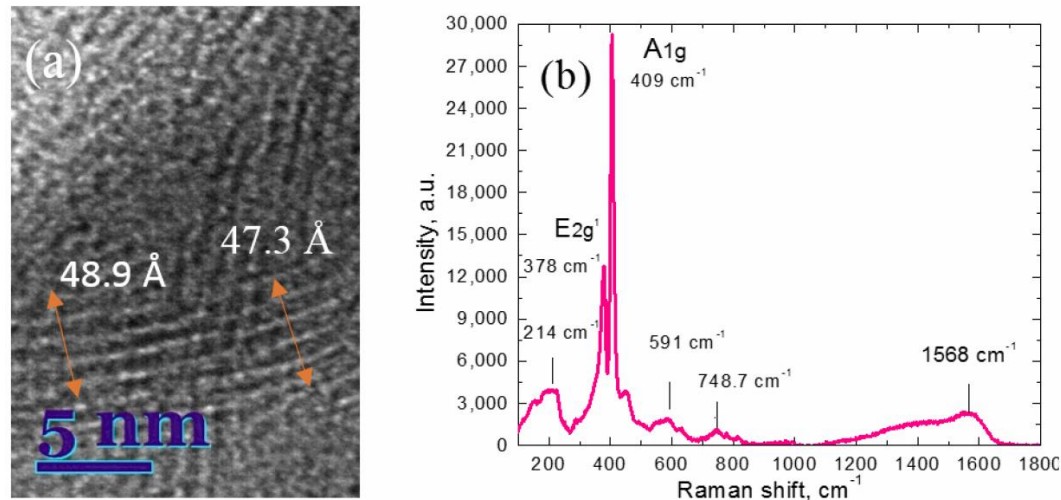

**Figure 2.** (**a**) HRTEM image and (**b**) Raman spectra of MoS$_2$ products synthesized in a solution containing 5 (NH$_4$)$_6$Mo$_7$O$_{24}$, 90 thiourea, and 25 mmol L$^{-1}$ of aniline at 220 °C for 10 h.

Apart from typical interlayer spacing of ~0.62 nm for pure MoS$_2$ [39], significantly larger interlayer spacing up to 0.98 nm, can be viewed. This, probably, occurred due to nonuniform intercalation of aniline molecules between S-Mo-S layers resulting in the nonuniform interlayer expansion. The addition of aniline to the synthesis solution was grounded on our recent findings that aniline affected on the activity of nanoplatelet-shaped MoS$_2$ hybrid electrocatalysts. As a result, twice more efficient and significantly stable catalysts for HER were designed upon hybridization with aniline. This effect was verified herein by a continuous potential cycling in the acidic solution within a range of −0.35 to 0.05 V vs. RHE potential containing 5 (NH$_4$)$_6$Mo$_7$O$_{24}$, 90 thiourea, and 25 mmol L$^{-1}$ of aniline at 220 °C for 10 h. We attributed this performance to a strong chemical coupling between the inserted aniline molecules and MoS$_2$ sheets, endowing them with more exposed active sites due to the lattice distortion and numerous defects viewed from the TEM observations (Figure 2a). Note that intercalation strategy of aniline and poly(aniline) molecules has been previously demonstrated for other HER catalysts with the lamellar structure allowing to enhance its activity [40,41]. The distorted MoS$_2$ lattice arrangements and defect-rich composition of the as-grown nanoplatelets at 220 °C for 10 h were also verified by Raman spectroscopy (Figure 2b) demonstrating clearly resolved A$_{1g}$ and E$_{2g}^1$ modes, centered at 409 and 378 cm$^{-1}$, respectively, of the predominant MoS$_2$ phases. The difference between the E$_{2g}^1$ and

A$_{1g}$ modes (31 cm$^{-1}$) implies that the structure of the MoS$_2$ in the nanocomposite consists of multiple layers [42]. Moreover, several additional modes, particularly a broad shoulder in a range from 1200 to 1650 cm$^{-1}$, centered at the 1568 cm$^{-1}$, implies that this mode should be ascribed to the vibration of intercalated aniline fragments [43]. Note, this band is absent in the Raman spectrum of the same MoS$_2$ synthesized without aniline (not showed here). At the same time, other less intense Raman signals, namely, at 214, 591, and 749 cm$^{-1}$ are observed, that cannot be attributed to molybdenum oxides.

As illustrated in Figure 3a–c, the morphology of products hydrothermally synthesized in the adapted herein solution at 220 °C for 5 h depends on the substrate material. Brussel cabbage-shaped balls comprised from the thin nanoplatelets were formed in the solution bulk and at the silica wafer substrate (Figure 3a), meanwhile, nanoplatelet of "grass" shape usually coated the metals, metal oxides, and even Teflon substrate (Figure 3b). Furthermore, in this study, the surprising effect was determined for hydrothermal formation of MoS$_2$ films onto the aluminum substrate. In this case, the formation of region at the metal substrate in an average thickness of ~0.42 μm composed of aluminum oxide nanowire arrays has been observed for the first time (Figure 3c).

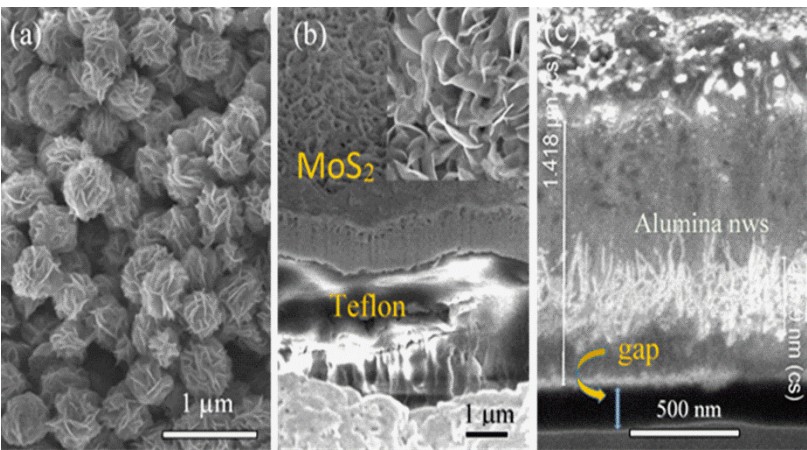

**Figure 3.** Top-side (**a**) and cross-sectional (**b**,**c**) SEM views of MoS$_2$ films fabricated onto the Si/SiO$_2$ (**a**), Teflon (**b**), and Al (**c**) substrates via hydrothermal synthesis in the solution containing (in mmol L$^{-1}$): 5 ammonium heptamolybdate, 90 thiourea, and 25 aniline at 220 °C for 10 h.

Figure 4 displays the Raman spectra of molybdenite films fabricated under the same autoclaving conditions on the molybdenum and titanium substrates as well as in the solution bulk clearly demonstrating an obvious compositional difference of the products formed. From these spectra, considerably pure MoS$_2$ species with clearly resolved characteristic phonon modes A$_{1g}$ at 409 cm$^{-1}$ and E$'_{2g}$ at 385 cm$^{-1}$ are formed in the solution bulk (Figure 4c) compared with the films grown at the substrates (Figure 4a,b). Furthermore, in accordance with the literature [42,44], the difference between E$'_{2g}$ and A$_{1g}$ peaks (24 cm$^{-1}$) strongly supports that MoS$_2$ cabbage-shape species are just few layered. In contrast, the shape of A$_{1g}$ and E$'_{2g}$ phonon modes as well as the large difference between E$'_{2g}$ and A$_{1g}$ peaks (>90 cm$^{-1}$) indicated on the significant degradation of the MoS$_2$ nanoplatelet material.

Figure 5 depicts the top-side (a) and cross-sectional (b) SEM images of MoS$_2$-based film fabricated onto the Ti substrate. These images outline that as-grown films are of nanosheet morphology with lateral size in the range of 100–200 nm. From the cross-sectional image (Figure 5b), an obvious gap between the film and substrate was, however, observed, which, of the most time, disfavors the charge transfer at the interface as well as a considerable decrease in the HER activity and durability even during a short HER processing (Figure 5c). In addition, the hydrogen evolution reaction at this electrode proceeds with trice larger Tafel slope compared with that at Pt electrode (Inset in Figure 5c). Better adhesion of MoS$_2$ film to substrate was not achieved by modulating the synthesis time within 2–15 h range. Upon an intense hydrogen gas evolution in an acidic solution, this film was destroyed quickly due to film sputtering from the Ti substrate resulting in the catalytic activity decrease (Figure 5c).

The gap between the Ti and $MoS_2$ film cannot be explained by a work function value of this metal (4.3 eV) [45]. We suspect that this effect could be ascribed to the formation of sublayer from $Ti(OH)_x$ due to the hydrothermal synthesis at 220 °C creating 23 bar pressure in the reactor and protecting a good adhesion of $MoS_2$ nanoplatelets.

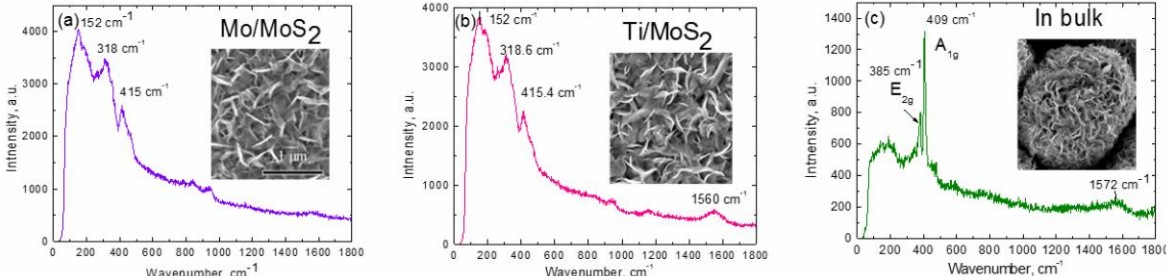

**Figure 4.** Raman spectra of the synthesized products on the Mo (**a**) and Ti (**b**) substrates, and in the solution bulk (**c**) by autoclaving in the solution containing 5 ammonium heptamolybdate, 90 thiourea and 25 mmol $L^{-1}$ aniline at 220 °C for 5 h. $\lambda_{exc}$ = 532 nm. In the insets, the corresponding SEM views of the resulted products.

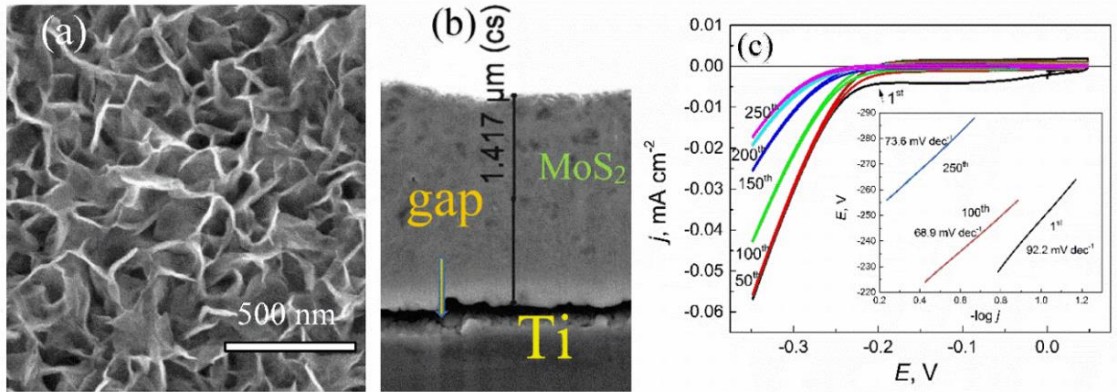

**Figure 5.** Top-side (**a**) and cross-sectional (**b**) SEM images of $MoS_2$ film formed onto the Ti substrate by hydrothermal treatment in the solution containing 5 ammonium heptamolybdate, 90 thiourea, and 25 mmol $L^{-1}$ aniline. In (**c**), the cycling voltammograms of this catalyst in the acidic solution at 10 mV $s^{-1}$ potential scan rate for indicated potential scan cycle are outlined. The inset outlined the Tafel slopes calculated for indicated potential scan.

From the SEM observations, the morphology of hybrid-type $MoS_2$ film formed on the molybdenum substrate under conditions of this study is quite similar to the one at the Ti substrate (Figure 6a,b). In sharp contrast, the nanoplatelets of this film are well attached to the Mo substrate (Figure 6c). The good adhesion of $MoS_2$ film to the Mo substrate can be attributed to a low work function (WF) of Mo (4.5 eV) because according to the Schottky Mott model, metal–semiconductor contacts would be based on the WF of metal [45]. To assess durability of this catalyst, we cycled them for 2000 HER cycles. A set of polarization curves obtained is shown in Figure 6d. From these curves, however, quite high activity for this electrocatalyst was determined only at the initial cycling. Upon 500–600 potential scans within 0.05 to −0.35 V range the catalytic activity of such electrocatalyst decreased in times, although the Tafel slope of electrochemical reaction changed insignificantly. For comparison, the HER activity at the Pt electrode under the same conditions attained ~100 mA $cm^{-2}$ at 0.2 V and proceeded with the Tafel slope of about 31 mV $dec^{-1}$ as is typical [46].

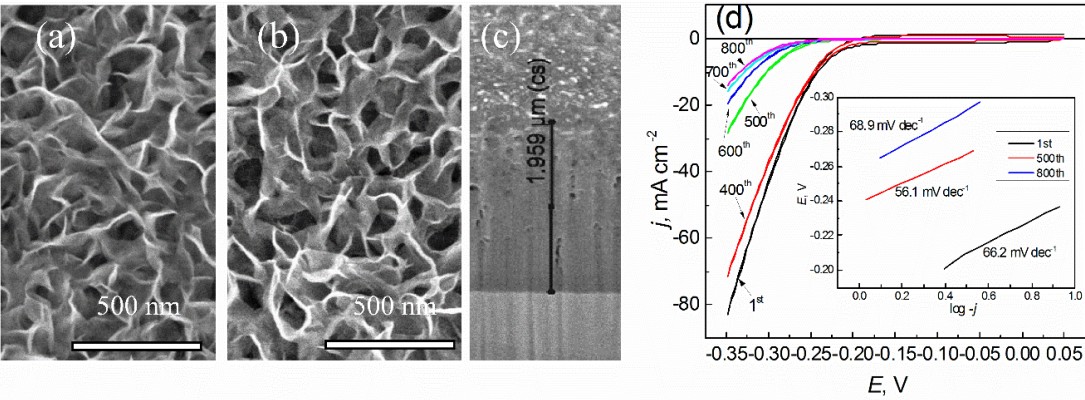

**Figure 6.** Top-side (**a**,**b**) and cross-sectional (**c**) SEM images of MoS$_2$ film formed onto the Mo substrate by hydrothermal treatment in the solution containing 5 ammonium heptamolybdate, 90 thiourea and 25 mmol L$^{-1}$ aniline at 220 °C for 5 h before (**a**) and after (**b**) 1000 potential cycling within −0.35 to 0.05 V potential range at 10 mV s$^{-1}$ rate. In (**d**), the set of HER voltammograms and the Tafel slopes calculated for indicated potential scan are outlined.

As the MoS$_2$-based films at the Ti substrate were unstable over a long period of the HER processing, we followed their stabilities after deposition onto the anodized Ti substrate. In this way, well-known Ti anodizing process [47] in the acidic aqueous solution containing 2.0 mol L$^{-1}$ H$_3$PO$_4$ and 0.2 mol L$^{-1}$ NH$_4$F at 20 V was chosen to design the typical nanotube-shaped film (Figure 7a).

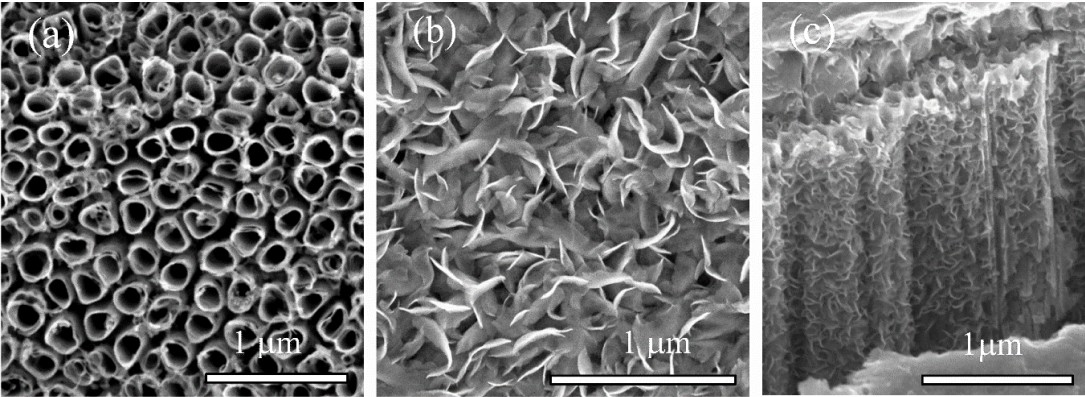

**Figure 7.** Top-side (**a**,**b**) and cross-sectional (**c**) SEM images of anodized Ti specimen before (**a**) and after (**b**,**c**) covering with MoS$_2$ film as in Figure 1.

Considering surprisingly low electrical resistance of the crystalline anatase-TiO$_2$ in the hydrogen gas environments [48], the anodized specimens were annealed and just then subjected to the hydrothermal treatment in the adapted herein solution under the identical conditions. The resulting film also demonstrated nanoplatelet's morphology (Figure 7b) and nice adhesion to the titania nanotubes (Figure 7c). Worth noticing that apart deposition onto the surface, MoS$_2$ species tightly filled titania tubes, cracks, and intertube gaps (Figure 7c), likely due to a strong chemical attachment to the TiO$_2$ surface [49]. As such, this electrode exhibited a high HER activity in the acidic solution (Figure 8a). Besides, this catalyst possessed exceptionally high HER stability even at 50–60 mA cm$^{-2}$ current density and a small Tafel slope decreasing from 79 to 66 mV dec$^{-1}$ during long period of exploitation (Figure 8a, inset). Note that for HER in acidic electrolyte, the theoretical Tafel slopes are 120, 40, and 30 mV dec$^{-1}$, corresponding to the Volmer, Heyrovsky, and Tafel step, respectively. A Tafel slope of 66 mV dec$^{-1}$ indicated that hydrogen evolution occurred via a fast discharge reaction (H$_3$O$^+$ + e$^-$ + cat = cat-H + H$_2$O) and thereafter, a rate determining (ion + atom) reaction (H$_3$O$^+$ + e$^-$ + cat-H = cat + H$_2$ + H$_2$O), that is, the Volmer–Heyrovsky mechanism [50]. Electrolytic

property robustness was confirmed by more than 15 h continuous HER processing in an acidic solution. The good catalytic activity and stability of MoS$_2$ films hybridized with aniline at the Ti/TiO$_2$ substrate was also confirmed by invariable values of the A$_{1g}$ (410 cm$^{-1}$) and E$_{2g}^1$ (383 cm$^{-1}$) vibrational modes characteristic to MoS$_2$ Raman spectra [51] (Figure 8b) and could be explained not only by the nanotubed morphology and chemical affinity of MoS$_2$ to titania surface as has been reported [48,51] but also by the low Ti/TiO$_2$ interface resistance occurring via hydrogen spilt over effect [52,53].

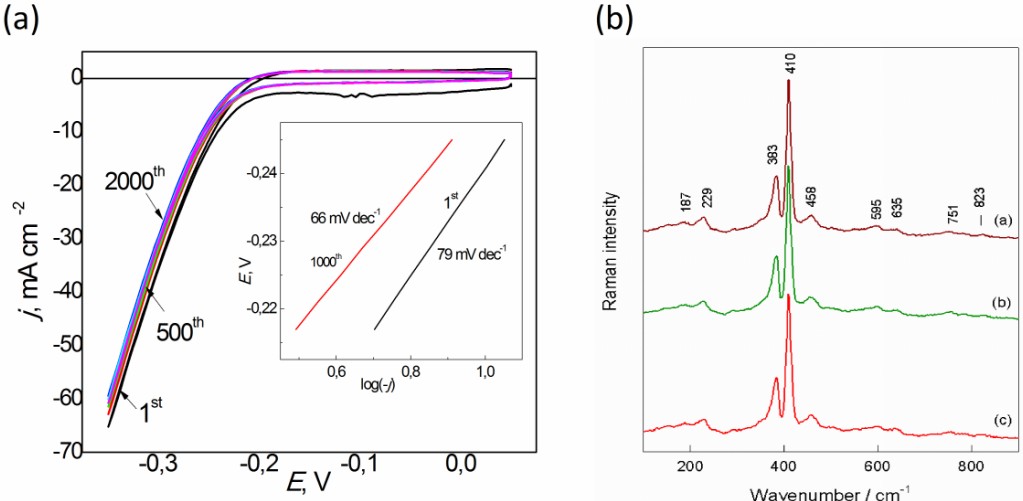

**Figure 8.** (**a**): HER voltammograms at the indicated potential scan number. (**b**) The Raman spectra of the samples after 1st (**a**), 500th (**b**), and 2000th potential scans. In the inset, the Tafel plots for the samples after 1st and 1000th potential scans are shown.

Titanium substrates covered with nanostructured titanium monoxide, particularly TiO$_x$ nanofeatures, were also tested. These nanostructured films (Figure 9a) were fabricated hydrothermally in the selenious acid solution as reported recently [37]. The quality of MoS$_2$ nanoplatelet array formed onto the Ti/TiO$_x$ substrate was dependent on the MoS$_2$ synthesis conditions. Thus, when Ti/TiO$_x$ electrode was hydrothermally treated in the heptamolybdate-thiourea solution at 220 °C for 2 h, just random depositions of MoS$_2$ species were obtained, whereas if syntheses were conducted for 5 h or longer, Ti/TiO$_x$ electrode were entirely covered with the nanoplatelet-shaped species (Figure 9b).

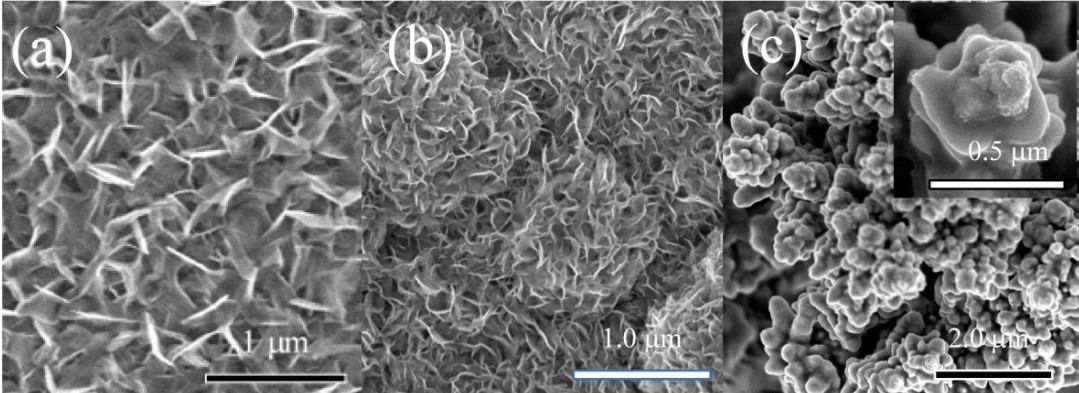

**Figure 9.** Top-side SEM views of TiO$_x$ (**a**) and TiO$_x$/MoS$_2$ film (**b**) surfaces before (**b**) and after (**c**) potential cycling in the solution of 0.5 H$_2$SO$_4$ + 0.25 mol L$^{-1}$ formic acid at 10 mV s$^{-1}$ within −0.45 to 0.05 V potential range for a period of 1000 potential scans. MoS$_2$ film was synthesized as in Figure 6.

Electrochemical tests in the acidic solution showed that H$_2$ evolution reaction at the surface of this film proceeds initially from the onset potential of ~170 mV vs. RHE (Figure 10) that can be

linked with the higher electrical conductivity of titanium monoxides comparing to $TiO_2$. However, the activity of this electrocatalyst decreases during prolonged potential cycling attaining 50–45 mA $cm^{-2}$ after 1000 cycles. Moreover, the pulsation of j(E) plots that was not characteristic for other substrates, was determined (Figure 10). We linked this effect with the polymerization of aniline molecules entrapped during $MoS_2$ hydrothermal synthesis and unhooked during the intense HER processing, covering $MoS_2$ platelets as clearly evidenced by the top-side film SEM image presented in Figure 9c.

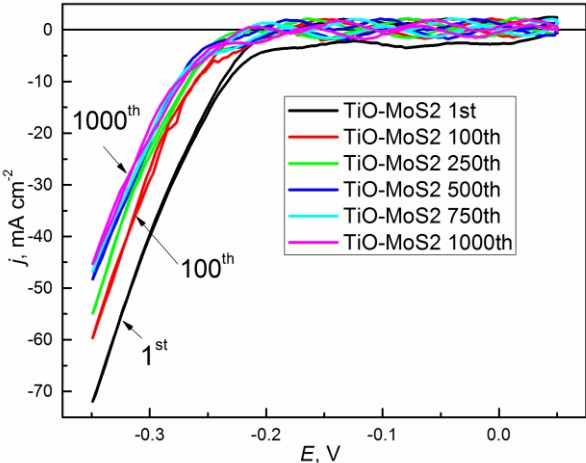

**Figure 10.** HER voltammograms at indicated potential scan number for the $Ti/TiO_x/MoS_2$-based electrode in the 0.5 $H_2SO_4$ and 0.25 mol $L^{-1}$ COOH solution at 10 mV $s^{-1}$ potential scan rate.

## 4. Conclusions

The influence of substrate material on the morphology, composition, and catalytic properties of the $MoS_2$ nanoplatelet-shaped films for hydrogen evolution reaction from the acidic solution was studied. One-pot hydrothermal synthesis approach was adapted as a simple and cost-effective way for the design of nanostructured catalyst films. To increase the HER activity of the synthesized material, aniline additives were successfully used for the first time. The compositional and processing optimization tests resulted in the formation of HER active 1–2 μm thick nanoplatelet $MoS_2$ film hybridized with aniline molecules as elucidated from the SEM view (Figure 7c) and Raman spectrum (Figure 2b), respectively. Via a long-term HER processing, it is determined that among various tested substrates, titanium covered by a thin nanotubed titania film is the most suitable for the preparation of well adherent to the substrate, efficient, and stably working catalyst.

**Author Contributions:** Data curation, S.R.; electrochemical investigations P.G.; structural analysis A.N. and M.K.; Raman analysis R.T.; writing—review and editing A.J. All authors have read and agreed to the published version of the manuscript.

**Funding:** P.G. acknowledged funds provided by the Lithuania Science Council via project no. 09.3.3-LMT-K-712-10-0039.

**Conflicts of Interest:** The authors declare no competing financial interest and any unpaid roles or relationships that might have a bearing on the publication process.

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
