# Peer review of "Substrate Impact on the Structure and Electrocatalyst Properties of Molybdenum Disulfide for HER from Water"

_metals, doi:10.3390/met10091251_

Round 1

Reviewer 1 Report

The authors present the impact of substrate treatment on the activity of MoS2 nanostructures in the electrocatalytic hydrogen evolution reaction. Although, the structures show good performance, some missing details results in objection of this reviewer.

  1. Abstract: The abstract is poor and not attractive to the readers. English editing needed (i.e. "pore stability). 
  2. Introduction: line 28: "...28 application of various new nanostructured materials are currently of high interest [2-4]." This is only partly true as there are other emerging materials for electrocatalytic HER, such as alloys (Metals 20188(2), 83), metallopolymers (Polymers 201911(1), 110; ACS Sustainable Chem. Eng. 2017, 5, 11, 10206–10214) and porous metals (Nanoscale 2017,9, 12231-12247). The introduction needs to be broadened. 
  3. Line 79: sub-section "Ti surface pre-treatment, anodizing and oxidation": This part needs clarification. What is the exact electrochemical set-up? What is the anode what is the cathode? Did the authors use the Ti foil as anode and two Pt foils as cathode? In this part, XPS surface analysis would be the logic consequence to exclude "leached Pt" as active species in HER. XPS surface analysis is needed to detect any possible Pt deposited on the MoS2 electrocatalyst. 
  4. What is the role of aniline in this study and what happens to it during thermal annealing?
  5. What is the reason that the authors of this study add formic acid to the 0.5 M H2SO4 electrolyte solution, which is not a common procedure in the literature?
  6. Line 185, Fig.5: "The Inset outlined the Tafel slopes calculated". whatever it means, there is not inset in Fig.5. 
  7. Extensive english editing needed. 

Reviewer 2 Report

It's an interesting paper about the application of MoS2 as HER electrocatalyst. The discussion of the substrate effect on the MoS2 HER activity is useful for the HER electrocatalyst development in the future. It can be considered to be published after the following issues are appropriately addressed. 

  1. Line 105-114: Did you keep nitrogen or hydrogen purging during the HER measurement? Since the counter electrode was in the same solution, the counter electrode reaction, OER, may affect your working electrode without protective gas purging. 
  2. Line 158-165: Please provide more discussion for the reason that different substrates can give out different morphologies. 
  3. Figure 5: Any reason for the gap between the MoS2 and Ti metal substrate? There is no such kind of gap between MoS2 and Teflon which is believed to have a lower affinity with the metal sulfide. 
  4. Please plot the HER activity of Pt electrode as the comparison. 
  5. Line 261: the high electrical conductivity or resistance? It looks like you want to say high electrical resistance. 
  6. Obviously, the substrates except the TiOx are flat. On contrast, TiOx is a tube-like substrate. And TiOx has the best effect on the loaded MoS2 in the durability and efficiency of HER.  Is it due to the unique morphology of TiOx substrate or its chemical affinity between TiOx and MoS2. 
  7. Extensive English editing is required. A lot of grammar mistakes, like line 59.

Reviewer 3 Report

Reviewing of the manuscript: Substrate impact on the structure and electrocatalyst properties of molybdenun  disulphides for HER from water.

The document describes the use of MoS2-based electrocatalysts for the hydrogen production from water.

The approach of  tailoring the morphology of the electrocatalytic material for enhancing their performance it is interesting. In addition, the different synthesis procedures applied seem to be successful for obtaining the desired effect in the prepared materials. This was clearly demonstrated by means of the different characterisation techniques.

The results are interesting and the materials promising, however the main weakness of the manuscript is the absence of comparison with other type of electrocatalytic materials (some reference systems). It has to be considered that the authors proposed their approach as a  next step in the design of the electrocatalysts based on the tailoring of the morphology, but references of “traditional” electrodes are not included. In fact, only ten references are included in the result section. So a deeper discussion, considering reference systems is very important to see the advantages of the obtained materials.

Regarding the level of English, even though I am not expert, I do strongly recommend a revision of the entire document by a native speaker. Many mistakes were detected along the manuscript.

Considering that described above, I recommend the publication of the manuscript only after major revisions.

Round 2

Reviewer 1 Report

The authors missed to do systematic surface analysis via XPS of the modified electrocatalysts, however I accept this manuscript for publishing in this journal after extensive English editing. There are still serious grammatical errors in this manuscript, which makes quite difficult for the reader to follow. 

Reviewer 2 Report

It's good to be published after the revision. 

Reviewer 3 Report

After reviewing the revised version of the manuscript “Substrate Impact on the Structure and Electrocatalyst 2 Properties of Molybdenum Disulfide for HER from 3 Water”, the improvement of the document it is observable. Therefore, I consider the authors have modified the manuscript attending the observations made by the reviewers.

In this sense, I recommend the publication of the manuscript in its present form.